# Evaluation of renal markers in systemic autoimmune diseases

**Hari Krishnamurthy** [ID]*, **Yuanyuan Yang, Qi Song, Karthik Krishna, Vasanth Jayaraman, Tianhao Wang, Kang Bei, John J. Rajasekaran**

Vibrant America LLC., San Carlos, CA, United States of America

* hari@vibrantsci.com

**Editor:** Alessandro Granito, University Hospital of Bologna Sant'Orsola-Malpighi Polyclinic Department of Digestive System: Azienda Ospedaliero-Universitaria di Bologna Policlinico Sant'Orsola-Malpighi Dipartimento dell'apparato digerente, ITALY

## Abstract

Renal involvement is a common occurrence in subjects with systemic autoimmune diseases. The renal manifestation and its severity depend on the underlying condition and may reversely complicate the clinical course of autoimmune diseases. Renal function markers have been widely used in the assessment of normal functioning of kidneys including glomerular filtration rate and concentrating and diluting capacity of the kidney. An increase or decrease in the values of these markers may indicate kidney dysfunction. In this study, a number of critical renal markers were examined in seropositive autoimmune diseases including systemic lupus erythematosus (SLE), connective tissue disorder (CTD), and rheumatoid arthritis (RA). The data from three cohorts of subjects enrolled in renal markers and autoimmune antibody testing between January 2015 to August 2019 were retrospectively studied. The prevalence of renal markers that were out of the reference range and their average levels in female and male subgroups across SLE, CTD, and RA cohorts were compared and analyzed. The levels of renal markers are significantly affected by the presence of autoantibodies, in particular eGFR, cystatin C, and albumin. Autoantibodies were also more frequent in subjects with severe renal function damage. Close follow-up of both renal markers and autoantibodies may potentially assist in the early diagnosis of kidney diseases and improve the survival and life expectancy of autoimmune patients.

## Introduction

Systemic autoimmune diseases are a large and heterogeneous group of immunologically mediated disorders that originated from complex genetic and environmental factors and are characterized by the production of autoantibodies [1, 2]. Renal involvement is a common occurrence in subjects with systemic autoimmune diseases [3]. The renal manifestation and its severity depend on the underlying disease and may reversely complicate the clinical course of autoimmune diseases [4]. As observed in previous studies, impairment of renal functions may occur in a variable prevalence in different systemic autoimmune diseases, such as approximately 50% in systemic lupus erythematosus (SLE) [5], roughly 5% in systemic scleroderma (SSc) [6], a variable occurrence in Sjögren syndrome [7, 8], and rare manifestation in rheumatoid arthritis (RA) [9]. For most systemic autoimmune diseases, renal involvement can be of significant prognostic value and often warrants specific immunosuppressive treatment [10]. Thus, it is important to diagnose and manage them at an early stage.

**Data Availability Statement:** All relevant data are within the paper. Any additional data will be available upon request from Vibrant America LLC by sending an email to our bioinformatics team at bioinformatics@vibrant-america.com or hari@vibrantsci.com.

**Funding:** Vibrant America provided funding for this study in the form of salaries for authors [YY, QS, KK, VJ, TW, KB, HK, JJR]. The specific roles of these authors are articulated in the 'author contributions' section. The funders had no role in study design, data collection and analysis, decision to publish, or preparation of the manuscript.

**Competing interests:** The authors have read the journal's policy and the authors of this manuscript have the following competing interests: YY and QS are paid employees of Vibrant America LLC. KK, VJ, TW, KB, HK, JJR, are paid employees of Vibrant Sciences LLC. Vibrant Sciences or Vibrant America could benefit from increased testing based on the results. There are no patents, products in development, or marketed products to declare. This does not alter our adherence to PLOS ONE policies on sharing data and materials.

A variety of methods are available to assist clinicians to assess renal functions and injuries. The estimated glomerular filtration rate (eGFR) is generally regarded as the most important measure of overall renal function [11]. Decreased GFR is generally accompanied by other renal functional variables. Urea is the waste product of protein metabolism and should be almost eliminated through urinary excretion. Blood urea nitrogen (BUN) quantifies the accumulation of urea in the blood and has been widely used to assess renal functions as well as cardiovascular diseases [12]. Serum creatinine supplemented BUN for renal assessment since mid-1900s and remains a laboratory parameter to estimate GFR [13]. Serum creatinine is the product of the nonenzymatic dehydration of muscle creatine and is usually formed at a relatively constant rate. Creatinine can be freely filtered by the glomerulus and not reabsorbed by the renal tubules. To improve the use of serum creatinine to estimate GFR, several serum creatinine-based equations have been developed [14]. Serum cystatin C, a low molecular weight protein in the cysteine proteinase inhibitor family, is another marker that has been considered enthusiastically to estimate GFR [15]. Unlike creatinine, serum cystatin C concentration appears to be independent of age, sex, and muscle mass. Beyond GFR, Albumin is one of the most prognostically significant biomarkers of kidney disease outcomes and even cardiovascular disease and death [16]. Electrolytes and minerals are frequently used to screen for an electrolyte or acid-based imbalance which may affect bodily organ function [17].

The purpose of this study was to evaluate a comprehensive panel of renal markers among seropositive autoimmune patients and seronegative controls. The renal function panel measured two critical calculated parameters and 11 markers which have been widely employed in clinical practice to monitor the physiologic status of the kidney. Three cohorts of patients were enclosed in this study and they were tested serologically for SLE, RA, CTD, and renal markers respectively. We have further attempted to analyze the renal markers' levels by females and males in relation to reference ranges. The frequency of autoantibodies across all three cohorts was investigated based on different stages of renal function damage.

## Materials and methods

### Serum samples

Three cohorts of retrospective study samples were included in this study, as shown in Table 1. The samples' medical information was collected between January 2015 to August 2019 and tested in the Vibrant America Clinical Laboratory (San Carlos, CA, USA). The waiver of consent for In Vitro Diagnostic Device study using leftover human specimens that are not individually identifiable was approved by the Western Institutional Review Board (WIRB) (work order #1-1098539-1).

### Renal function panel

The Renal function panel included electrolytes (sodium, potassium, chloride, total bicarbonate), minerals (calcium, phosphorus, magnesium), protein (albumin, cystatin C), waste products (BUN, creatinine), and two calculated values (BUN/creatinine ratio, estimated glomerular filtration rate (eGFR)). These markers were quantitatively determined at Vibrant America Clinical Laboratory (San Carlos, CA, USA), a CLIA-certified clinical laboratory. Detailed

**Table 1. Demographics of the three cohorts in this study.**

|  | Cohort 1 | Cohort 2 | Cohort 3 |
|---|---|---|---|
| Panels tested | SLE + Renal Function | CTD + Renal Function | RA + Renal Function |
| Number of subjects | 13841 | 9995 | 20681 |
| Genders | 8954 F / 4887 M | 6293 F / 3702 M | 13482 F / 7199 M |
| Average age (±SD) | 46 (±16) | 46 (±16) | 47 (±16) |

information regarding the markers' measurement is in **S1 File**. eGFR was calculated from serum creatinine using the CKD-EPI equation for adults (> 18 years old) and Bedside IDMS-traceable Schwartz GFR Calculator for Children (≤ 18 years old). The reference range for normal levels of renal markers is detailed in **S2 File**.

## Systemic Lupus Erythematosus (SLE) panel

The SLE panel included antinuclear antibody (ANA) and anti-dsDNA antibody. The ANA detection was performed with a solid phase bio-chip immunofluorescence assay, Vibrant™ ANA HEp-2 (Vibrant America, LLC, San Carlos, CA, USA). A sample was considered ANA positive (ANA+) if any specific staining (homogeneous, centromere, speckled, nucleolar, peripheral) was observed to be greater than the negative controls. 1:40 dilution was used for screening and was reflexed depending on the assay results (1:80, 1:160, 1:320, 1:640, and so on). The elderly, especially women, are prone to develop low-tittered autoantibodies in the absence of clinical autoimmune disease. Anti-dsDNA antibody was detected using a solid phase bio-chip immunofluorescence assay that reports qualitative and semi-quantitative results. A seropositive SLE subject is whose ANA and anti-dsDNA testing results were both positive. A seronegative control is ANA and/or anti-dsDNA antibody were negative.

## Connective Tissue Disorder (CTD) panel

The CTD panel included ANA and 10 anti-extractable nuclear antigens (ENA). The testing principles and assay process of detecting ANA and 10 anti-ENA were very similar to the procedures described in our previous work [18]. The 10 anti-ENA antibodies including SSA(Ro), SSB(La), RNP/Sm, Jo-1, Sm, Scl-70, Chromatin, Centromere, Histone, RNA polymerase III was tested. SSA(Ro), SSB(La), RNP/Sm, and Jo-1 were detected using a solid phase bio-chip immunofluorescence assay that reports qualitative and semi-quantitative results. The assessment and interpretation of the results were following the international guideline announced by the European autoimmunity standardization initiative and the International Union of Immunologic Societies/World Health Organization/Arthritis Foundation/Centers for Disease Control and Prevention autoantibody standardizing committee. A seropositive CTD subject is whose ANA and more than one of the anti-ENAs testing results were positive. A seronegative control is whose ANA and/or anti-ENAs testing results were negative.

## Rheumatoid Arthritis (RA) panel

The RA panel included anti-RF IgM (Roche Diagnostics, Risch-Rotkreuz, Switzerland) and anti-CCP3 IgG and IgA (Inova Diagnostics, San Diego, CA, USA). The interpretation of the results strictly followed the protocol suggested by the assay provider companies. A seropositive RA subject is someone who has at least one antibody at borderline of or more than an index value of 0.95. A seronegative control is if the concentrations of the antibodies to all the markers in the panel were equal to or less than the cut-off values.

## Data analysis

Clinical data from the de-identified subjects were included in a database retrieved through MySQL workbench 8.0.12 and analyzed using R for Windows version 3.5.1. Two-tail student T test was performed to determine whether there is significant difference between data sets and $P<0.05$ is considered as significant. In all histogram figures, P values less than 0.05 were given *, P values less than 0.01 were given **, P values less than 0.001 were given ***, P values less than 0.0001 were given ****, and P values more than 0.05 were not labeled.

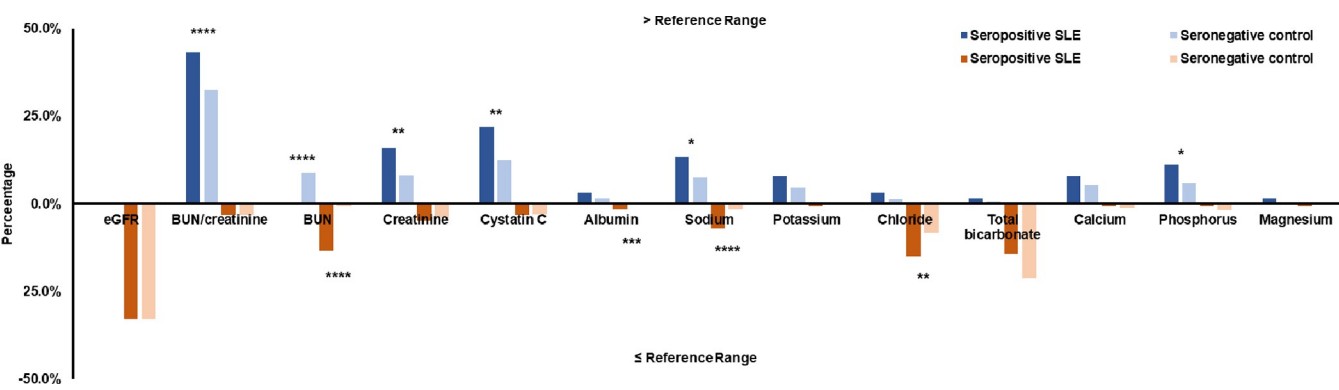

**Fig 1. Prevalence of out-of-range renal markers in seropositive SLE subjects and seronegative controls.**

## Results

### Renal makers in SLE subjects

Cohort 1 consists of 126 seropositive SLE subjects and 13715 seronegative controls. Fig 1 shows the prevalence of SLE and control subjects for carrying higher-than-reference-range (blue, above axis) and lower-than-reference-range renal markers (orange, below axis). Among them, 43.4% of SLE subjects had high BUN/creatinine levels compared with 32.5% of controls (P<0.0001). This result is, however, opposite to the observation with BUN, which were carried by more SLE subjects when lower than reference range (P<0.0001) and by more controls when higher than the reference range (P<0.0001). Low sodium level was also found to be significantly more frequent in SLE subjects (P<0.0001).

The results of subjects with higher-than-reference-range renal markers are shown in blue. The results of the subject with lower-than-reference-range renal markers are shown in orange.

We split this cohort into female and male subgroups and assessed their levels of renal markers, respectively, shown in Table 2. The mean eGFR level was lower in the female SLE subjects but not much different in the male group. Beyond eGFR, the mean values of BUN and cystatin

**Table 2. Average levels of renal markers among seropositive SLE subjects and seronegative controls.**

| Renal Markers (Unit) | Female | | | Male | | |
|---|---|---|---|---|---|---|
| | Seropositive SLE | Control | P value | Seropositive SLE | Control | P value |
| | Mean ± SD | Mean ± SD | | Mean ± SD | Mean ± SD | |
| eGFR | 82.7±22.2 | 90.5±20.8 | <0.001 | 80.3±27.8 | 85.2±18.7 | 0.415 |
| BUN/creatinine | 20.1±5.3 | 19.3±6.1 | 0.145 | 18.2±5.1 | 17.3±5.5 | 0.413 |
| BUN (mg/dL) | 15.6±4.9 | 13.9±4.7 | <0.00001 | 21.0±10 | 16.4±5.5 | 0.024 |
| Creatinine (Mg/Dl) | 0.8±0.2 | 0.7±0.2 | 0.129 | 1.1±0.4 | 1.0±0.2 | 0.081 |
| Cystatin C (mg/L) | 1.0±0.3 | 0.9±0.2 | <0.00001 | 1.2±0.4 | 1.0±0.3 | 0.006 |
| Albumin (G/Dl) | 4.4±0.4 | 4.6±0.3 | <0.001 | 4.5±0.4 | 4.6±0.3 | 0.138 |
| Sodium (Mmol/L) | 141.5±3.8 | 141.5±2.8 | 0.864 | 141.3±4.1 | 142.0±2.8 | 0.423 |
| Potassium (Mmol/L) | 4.4±0.4 | 4.4±0.4 | 0.799 | 4.9±1.2 | 4.5±0.4 | 0.068 |
| Chloride (Mmol/L) | 101.0±3.5 | 101.3±2.8 | 0.430 | 100.5±3.9 | 101.0±2.9 | 0.535 |
| Total bicarbonate (Mmol/L) | 22.5±3 | 22.3±2.9 | 0.603 | 22.3±3.4 | 22.7±3.1 | 0.497 |
| Calcium (Mg/Dl) | 9.8±0.4 | 9.6±0.4 | <0.001 | 9.6±0.4 | 9.7±0.4 | 0.407 |
| Phosphate (Mg/Dl) | 3.8±0.6 | 3.7±0.5 | 0.055 | 3.7±0.6 | 3.5±0.6 | <0.000001 |
| Magnesium (Mg/Dl) | 2.1±0.2 | 2.1±0.2 | 0.415 | 2.1±0.2 | 2.1±0.2 | 0.270 |

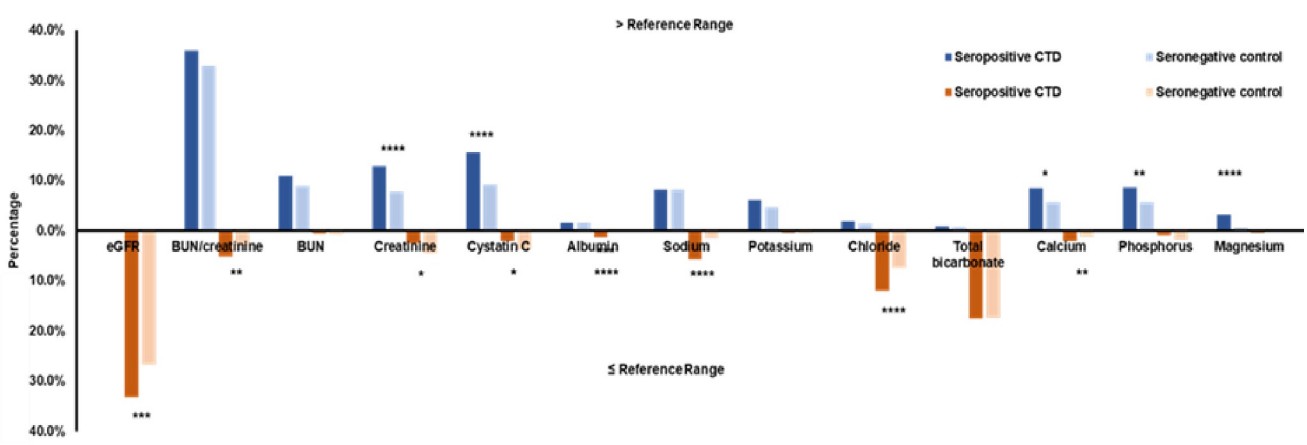

**Fig 2. Prevalence of out-of-range renal markers in seropositive CTD subjects and seronegative controls.**

C were higher in both female and male SLE subjects than respective controls. The mean calcium level was higher solely in SLE female subjects while phosphate level was higher solely in the SLE male subjects.

## Renal markers in CTD subjects

Cohort 2 consists of 695 seropositive CTD subjects and 9300 seronegative controls. The percentages of CTD and control subjects for carrying higher-than-reference-range (blue, above axis) and lower-than-reference-range renal markers (orange, below axis) are shown in Fig 2. In this cohort, 33.2% of the CTD subjects had low eGFR compared with 26.8% in controls (P<0.001). BUN/creatinine was also observed to be low in 5.1% of CTD subjects which is more frequent than 2.9% in controls (P<0.01). Other than these two calculated parameters, the higher-than-reference-range renal markers that were more frequent in CTD subjects include creatinine, cystatin C, calcium, phosphate, and magnesium. The lower-than-reference-range renal markers that were more frequent in CTD subjects include albumin, sodium, chloride, calcium.

The results of subjects with higher-than-reference-range renal markers are shown in blue; and the results of subjects with lower-than-reference-range renal markers are shown in orange.

Cohort 2 was split into female and male subgroups and the mean levels of renal marker in each group were shown in Table 3. The mean eGFR was significantly lower in both the female and male CTD subjects while there was no difference for BUN/creatinine. The mean levels of BUN, creatinine, cystatin C were higher in both female and male CTD subjects while albumin and total bicarbonate were higher in seronegative controls. The other markers did not show any statistical difference between seropositive and seronegative controls in this cohort.

## Renal markers in RA subjects

Cohort 3 consists of 3304 seropositive RA subjects and 17377 seronegative controls. Fig 3 demonstrates the prevalence of RA and control subjects with higher-than-reference-range (blue, above axis) or lower-than-reference-range renal markers (orange, below axis). Low eGFR and high BUN/Creatinine were more prominent in the seropositive group than the controls (35.6% vs. 23.3% for eGFR, 36.8% vs. 33.9% for BUN/Creatinine). Furthermore, high levels of BUN, creatinine, cystatin C, potassium, calcium, magnesium, and low levels of albumin, sodium, chloride were found to be more frequent among RA subjects (P<0.05).

**Table 3. Average levels of renal markers among seropositive CTD subjects and seronegative controls.**

| Renal Markers (Unit) | Female | | | Male | | |
|---|---|---|---|---|---|---|
| | Seropositive CTD | Control | P value | Seropositive CTD | Control | P value |
| | Mean ± SD | Mean ± SD | | Mean ± SD | Mean ± SD | |
| eGFR | 87.2±20.8 | 92.2±20.9 | <0.000001 | 76.5±21.9 | 83.7±18 | <0.001 |
| BUN/creatinine | 19.7±6.3 | 19.5±6.1 | 0.344 | 16.5±7.2 | 17.1±5.3 | 0.324 |
| BUN (mg/dL) | 14.4±4.9 | 13.8±4.7 | <0.01 | 18.9±7.8 | 16.4±5.4 | <0.001 |
| Creatinine (Mg/Dl) | 0.7±0.2 | 0.7±0.2 | 0.016 | 1.1±0.3 | 1.0±0.2 | 0.003 |
| Cystatin C (mg/L) | 0.9±0.2 | 0.9±0.2 | <0.000001 | 1.1±0.4 | 1.0±0.2 | <0.0001 |
| Albumin (G/Dl) | 4.5±0.3 | 4.6±0.3 | <0.00001 | 4.5±0.4 | 4.6±0.3 | <0.00001 |
| Sodium (Mmol/L) | 141.5±3.2 | 141.6±2.9 | 0.184 | 141.3±3.2 | 142.0±2.8 | 0.013 |
| Potassium (Mmol/L) | 4.4±0.4 | 4.4±0.4 | 0.229 | 4.5±0.7 | 4.5±0.4 | 0.506 |
| Chloride (Mmol/L) | 101.4±3 | 101.5±2.8 | 0.492 | 100.5±3.2 | 101.1±2.8 | 0.035 |
| Total Bicarbonate (Mmol/L) | 22.0±2.8 | 22.3±3 | 0.033 | 22.3±3.2 | 22.8±3.1 | 0.046 |
| Calcium (Mg/Dl) | 9.6±0.5 | 9.6±0.4 | 0.668 | 10.1±2 | 9.7±0.4 | 0.013 |
| Phosphate (Mg/Dl) | 3.8±0.5 | 3.7±0.5 | 0.090 | 4.4±3.1 | 3.5±0.6 | <0.001 |
| Magnesium (Mg/Dl) | 2.1±0.2 | 2.1±0.2 | 0.097 | 2.1±0.2 | 2.1±0.2 | 0.276 |

The results of subjects with higher-than-reference-range renal markers are shown in blue; and the results of subjects with lower-than-reference-range renal markers are shown in orange.

The mean levels of the renal markers in both female and males are presented in Table 4. eGFR was elevated in both the female and male RA subjects while BUN/Creatinine was lowered in the female RA subjects. BUN, creatinine, cystatin C were elevated in both gender groups. Several markers' average levels were greater in the control groups including albumin, sodium, chloride, calcium, and phosphate.

## Evaluation of autoantibodies at different stages of renal damage

In this study, we divided all three cohort subjects into three stages in terms of kidney damage based on their eGFR: 90 or above indicating normal GFR; 60 to 89 indicating mild decreased GFR; below 60 indicating moderate to severe decreased GFR. The frequency of 14 autoantibodies were examined by stages and displayed in Fig 4. There is an apparent trend that autoantibodies became more frequent as eGFR decreased, which indicates worse renal damage. ANA,

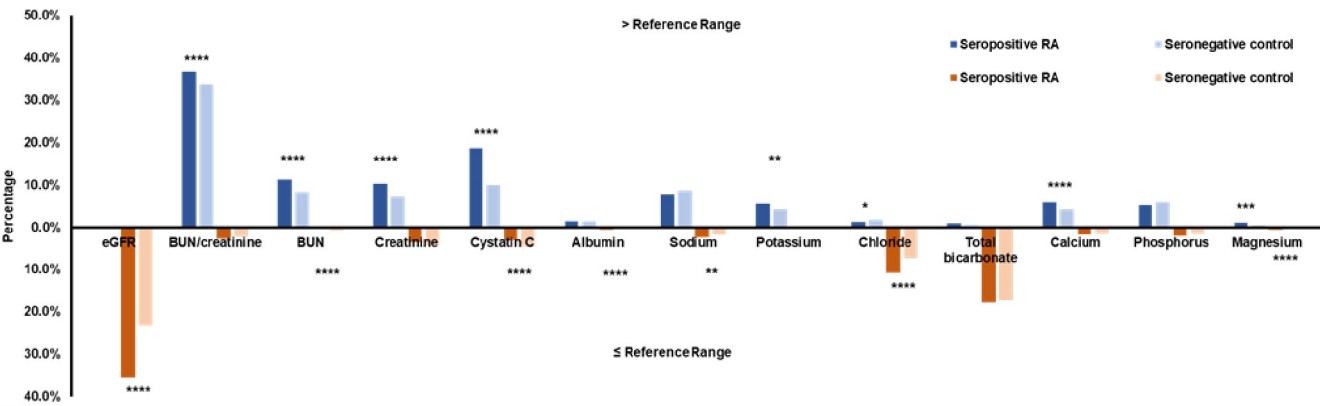

**Fig 3. Prevalence of out-of-range renal markers in seropositive RA subjects and seronegative controls.**

**Table 4. Average levels of renal markers among seropositive RA subjects and seronegative controls.**

| Renal Markers (Unit) | Female | | | Male | | |
|---|---|---|---|---|---|---|
| | Seropositive RA | Control | p value | Seropositive RA | Control | p value |
| | Mean ± SD | Mean ± SD | | Mean ± SD | Mean ± SD | |
| eGFR | 86.9±21 | 94.2±20.3 | <0.000001 | 80.5±17.8 | 84.7±17.6 | <0.000001 |
| BUN/creatinine | 20.3±6.5 | 19.5±6.1 | <0.000001 | 17.5±5.9 | 17.3±5.2 | 0.163 |
| BUN (mg/dL) | 14.8±5.1 | 13.8±4.7 | <0.000001 | 17.2±5.7 | 16.5±5.5 | <0.00001 |
| Creatinine (Mg/Dl) | 0.8±0.6 | 0.7±0.2 | <0.001 | 1.0±0.2 | 1.0±0.2 | 0.042 |
| Cystatin C (mg/L) | 0.9±0.3 | 0.9±0.2 | <0.000001 | 1.0±0.3 | 0.9±0.2 | <0.000001 |
| Albumin (G/Dl) | 4.5±0.3 | 4.6±0.3 | <0.000001 | 4.6±0.3 | 4.7±0.3 | <0.000001 |
| Sodium (Mmol/L) | 141.5±3 | 141.7±2.9 | 0.002 | 141.8±2.9 | 142.1±2.8 | 0.006 |
| Potassium (Mmol/L) | 4.4±0.4 | 4.4±0.4 | 0.461 | 4.5±0.4 | 4.5±0.4 | 0.053 |
| Chloride (Mmol/L) | 101.2±3 | 101.6±2.9 | <0.000001 | 100.9±3.1 | 101.2±2.8 | 0.002 |
| Total Bicarbonate (Mmol/L) | 22.5±3 | 22.3±2.9 | 0.004 | 22.6±3.1 | 22.7±3.1 | 0.313 |
| Calcium (Mg/Dl) | 9.6±0.5 | 9.6±0.4 | 0.002 | 9.6±0.5 | 9.7±0.4 | 0.034 |
| Phosphate (Mg/Dl) | 3.7±0.5 | 3.7±0.5 | 0.002 | 3.4±0.6 | 3.5±0.6 | <0.000001 |
| Magnesium (Mg/Dl) | 2.1±0.2 | 2.1±0.2 | 0.060 | 2.1±0.2 | 2.1±0.2 | 0.308 |

anti-histone, and anti-CCP were the three most prevalent autoantibodies among all renal damaged subjects and their presence were significantly more frequent than that in subjects with normal GFRs (P<0.05).

## Discussion

Patients with autoimmune diseases usually have increased mortality due to multiple factors that include an increased susceptibility to organ damage. The survival of patients with

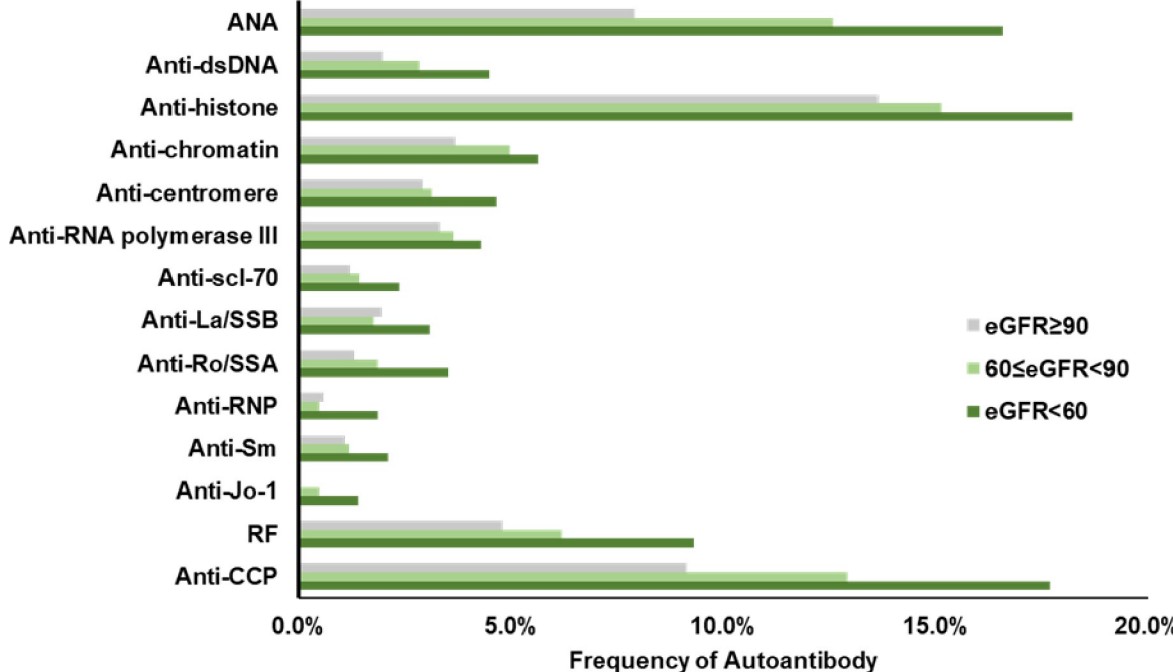

**Fig 4. Frequencies of individual autoantibodies in three stages of kidney diseases.** Autoantibodies increase with a decrease in eGFR.

autoimmune diseases has improved tremendously in the past few decades, which is attributed to early diagnosis, novel treatments, and better supportive care for organ failures. However, patients with autoimmune diseases still have a higher mortality rate than that of the general population. Renal disease is a major organ manifestation of autoimmune diseases, and it may lead to kidney failure in a proportion of patients over time. A number of previous studies have focused on the prevalence of specific renal dysfunctions among certain types of autoimmune diseases, but a comprehensive evaluation of clinically significant renal markers related with autoimmune serology conditions have not been investigated.

Kidney can be a frequent target in CTD because kidney has abundant connective tissues and active blood supplies. Patients with CTD-associated renal involvement are often asymptomatic, at least in an early disease course, or report nonspecific symptoms. Direct involvement of kidneys is usually less common in RA but it can be complication of therapy. Our study assessed a total of 13 renal function markers in three cohorts of subjects with positive/negative serology in SLE, CTD, or RA. We further divided each cohort into female and male subgroups and analyzed the mean levels of these renal markers. We observed significantly lower eGFRs in seropositive CTD and RA subjects compared with respective controls. In the cohort 2, more seropositive CTD subjects had eGFR lower than 60 (33.2% in CTD, 26.8% in control, P<0.001). The mean eGFR reduced to 5 when at least one of the anti-ENA markers appear in female subjects and the number becomes 7.2 with male subjects. A similar trend was also observed in cohort 3. More seropositive CTD subjects had eGFR lower than 60 (35.6% in RA, 23.3% in control, P<0.0001). The mean eGFR reduced 7.3 when either RF or anti-CCP appear in female subjects as compared with 4.2 in male subjects. It is also worthy to note that male subjects with positive autoimmune serology had the lowest average eGFR across all cohorts. These observations are in accordance with the co-existence of renal dysfunction and autoimmune diseases [19].

The other renal function markers also provided valuable correlation information. The BUN/creatinine ratio, as a differential marker of acute or chronic renal disease, is usually between 10:1 and 20:1 in healthy population but rises when the kidney blood flow decreases. In our study, the mean BUN/creatinine values were higher among females than males (P<0.05) but only the female RA subjects showed a statistical difference compared with controls (<0.000001). BUN and creatinine as individual markers were believed to have limited utilization in characterizing renal functions because they are easily interfered by increased dietary protein intake, hyper catabolism, corticosteroid use, or gastrointestinal bleeding. However, in this study, they displayed differences between seropositive groups and controls across all three cohorts except for creatinine in the SLE group. Cystatin C, as an emerging marker other than creatinine to calculate eGFR, did present a higher level of difference between the seropositive subjects and respective controls across all three cohorts. Similarly, higher mean levels of albumin were also prominently shown in majority of the seropositive subjects except for SLE males. eGFR, Cystatin C, BUN, Calcium, and Albumin showed a significant association with seropositive females in the SLE group. Lupus nephritis is a common manifestation of SLE. Its onset is between 3–5 years after SLE onset. Lupus nephritis is more common in women. This is consistent with our results. Regular monitoring of kidney function by measuring renal markers and preventing the decline of kidney function is the primary treatment of Lupus nephritis [20].

In terms of electrolytes and minerals, renal dysfunctions are often accompanied by elevations in potassium, phosphate, magnesium and decreases in sodium and calcium. The electrolytes and minerals are less specific to kidney, but they are a good indication for organ dysfunction induced chemical imbalance. Potassium is considered to be the most convincing electrolyte marker of renal failure and hyperkalemia, which is the most significant and life-

threatening complication of renal failure [21]. From our observation, there was no certain trend and the electrolytes are similar in seropositive subjects and respective controls with a few exceptions labeled in Figs 1–3. We hypothesized that the levels of these chemicals might be highly sensitive to diet and physical activity. These environmental factors may play an important role in regulating electrolytes and minerals.

Moreover, we observed a significantly greater prevalence of autoantibodies in more severe kidney damaged subjects. ANA, anti-histone, and anti-CCP were the three most prevalent autoantibodies in all renal damaged subjects and their presence were significantly more frequent than that in subjects with normal GFRs ($P<0.05$). Several mechanisms have been envisioned for renal involvement in systemic autoimmune diseases. Autoimmunity induced renal damage may be resulted from a systemic disturbance of immunity and accompanied by reduced tolerance to normal cellular and extracellular proteins [22]. Glomerular, tubular and vascular structures often become targets due to the loss of immunity balance. One hypothesis is that renal tissue may harbor self-antigens [23]. Autoantigens have been widely accepted as direct indicators in autoimmune disease; however, very few of them have been speculated to cause tissue injuries in kidney. Another theory is that the kidneys may become affected by the autoantigen outside the kidney [4]. The high flow, high-pressure perm-selective filtration function of the glomerulus may drive non-renal autoantigens to become renal targets during physiological process. Circulating autoantigens can accumulate in glomeruli and deposit as a target antigen because of their physio-chemical properties that predispose them to the glomerular structure. Moreover, antigen and antibodies may be neither derived nor deposited within the kidneys but the interaction between them may cause the disease [24]. The investigation of the mechanism based on the above speculations is currently underway. There were some limitations in our study that should be considered while reviewing the data. This was a retrospective study performed on de-identified serological data. The clinical conditions of the individuals were unknown and hence not part of the analysis. The testing was performed based on physician orders which would have an inherent bias on who was getting tested.

In conclusion, to the best of our knowledge this is the first retrospective study of patients with seropositive autoimmune diseases whose renal markers were examined and compared with seronegative controls. The renal markers reported in this paper are based on a large cohort of controlled samples and an extended list of autoantibodies. The clinical utility of testing for renal markers along with autoantibodies is immediately apparent and may potentially assist in early diagnosis of kidney diseases and improve survival and life expectancy of autoimmune patients.

## Supporting information

**S1 File. Measurement of renal functional panel markers.**
(DOCX)

**S2 File. Reference range for normal levels of renal markers.**
(DOCX)

## Author Contributions

**Conceptualization:** Hari Krishnamurthy, Vasanth Jayaraman, John J. Rajasekaran.

**Data curation:** Yuanyuan Yang, Qi Song.

**Formal analysis:** Yuanyuan Yang.

**Methodology:** Karthik Krishna, Tianhao Wang.

**Software:** Kang Bei.

**Visualization:** Hari Krishnamurthy, Vasanth Jayaraman, John J. Rajasekaran.

**Writing – original draft:** Hari Krishnamurthy, Yuanyuan Yang.

**Writing – review & editing:** Hari Krishnamurthy, Yuanyuan Yang, Qi Song, Karthik Krishna, Vasanth Jayaraman, Tianhao Wang, Kang Bei, John J. Rajasekaran.

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
