## [Decision Letter · Decision Letter 0]

19 Dec 2022

PONE-D-22-31509Evaluation of Renal Markers in Systemic Autoimmune DiseasesPLOS ONE

Dear Dr. Krishnamurthy,

Thank you for submitting your manuscript to PLOS ONE. After careful consideration, we feel that it has merit but does not fully meet PLOS ONE’s publication criteria as it currently stands. Therefore, we invite you to submit a revised version of the manuscript that addresses the points raised during the review process.

We look forward to receiving your revised manuscript.

Kind regards,

Latika Gupta

Academic Editor

PLOS ONE

Journal Requirements:

"The authors have read the journal’s policy and the authors of this manuscript have the following competing interests: YY and QS are paid employees of Vibrant America LLC. KK, VJ, TW, KB, HK, JJR, are paid employees of Vibrant Sciences LLC. Vibrant Sciences or Vibrant America is a commercial lab which performs serological testing for autoimmune antibodies and sex hormones. Vibrant Sciences or Vibrant America could benefit from increased testing based on the results. There are no patents, products in development, or marketed products to declare. This does not alter our adherence to PLOS ONE policies on sharing data and materials."

We note that you received funding from a commercial source: "Vibrant America LLC"

"We acknowledge Vibrant America LLC for supporting this research."

"Vibrant America provided funding for this study in the form of salaries for authors [YY, QS, KK, VJ, TW, KB, HK, JJR]. The specific roles of these authors are articulated in the ‘author contributions’ section. The funders had no role in study design, data collection and analysis, decision to publish, or preparation of the manuscript. "

Additional Editor Comments:

Please find appended reviewer comments for addresal.

Reviewers' comments:

Reviewer's Responses to Questions

**Comments to the Author**

1. Is the manuscript technically sound, and do the data support the conclusions?

Reviewer #1: Partly

Reviewer #2: Yes

2. Has the statistical analysis been performed appropriately and rigorously? 

Reviewer #1: N/A

Reviewer #2: Yes

3. Have the authors made all data underlying the findings in their manuscript fully available?

Reviewer #1: No

Reviewer #2: Yes

4. Is the manuscript presented in an intelligible fashion and written in standard English?

Reviewer #1: Yes

Reviewer #2: Yes

5. Review Comments to the Author

Reviewer #1: Summary and general comments:

In the manuscript “Evaluation of Renal Markers in Systemic Autoimmune Diseases” the authors took a correlation approach to investigate the association between several renal markers and kidney disease in patients with systemic autoimmune diseases. To achieve such aim, the authors compared retrospectively 13 renal markers together with autoantibodies in 3 cohorts of patients: 1) 126 with systemic lupus erythematosus (SLE), 2) 695 with connective tissue diseases (CTDs) and 3) 3304 with rheumatoid arthritis (RA).

The results showed that:

1. Renal markers were elevated in the 3 cohorts of patients comparatively to controls.

2. Some autoantibodies such ANA, anti-Histone and anti-CCP were associated with kidney damage and lower eGFRs.

3. All the renal markers except for electrolytes were good in assessing renal involvement in patients with autoimmune diseases.

Overall, the paper is well organized and I enjoyed reading it but there are some rectifications and explanations that need to be addressed.

Specific comments:

#1 In the text, the authors mention 4 tables that do not exist in the manuscript version submitted for review. Please provide the manuscript with the cited tables.

#2 The last paragraph of the introduction is a bit misleading. For instance, the authors mention 13841 SLE patients when in reality it is 126 patients and 13715 controls. And then who were these control subjects, and why did each of the 3 cohorts have its own control group?

#3 In the statistical analysis section, the authors did not specify the type of statistical tests used.

#4 In the results section, the authors did not detail the quantitative measurements of the different markers assessed in the patients. It would be more appropriate to compare the means or to perform non-parametric tests followed by ROC analysis to evaluate the sensitivity and specificity of the markers in detecting renal damage. Eventually, the authors could perform a logistic regression to estimate the incremental risk for each one-unit increase in each marker and build a predictive model for kidney damage.

#5 In SLE patients, the association of anti-DNA antibodies with renal damage has been known for a long time and numerous human and mouse studies have confirmed the pathogenic role of these autoantibodies. The authors should discuss the discrepancy of their results with the literature.

Reviewer #2: 1. About leftover samples: when, on an average, were the samples processed after being leftover and is there any effect on the sample results because of the leftover time.

2. The titre and the strength of ANA control is not mentioned anywhere in manuscript.

3. Renal markers in SLE: Among the 43.4% of SLE subjects, is there any clinical data about how many were having or developed nephritis or other renal complications, that way the results would be translated more effectively bedside. in various other studies, serum creatinine level remains statistically significant risk factor for developing nephritis in SLE patients, what about the serum creatinine levels in this study, was that statistically significant? is there any clinical data available about those who have low sodium levels because some studies have demonstrated low sodium level as an inflammatory marker of lupus, the current study would further add strength to that association in case. Was the mean Calcium level in female SLE subjects and high phosphate level in male SLE patients statistically significant? there are some studies which show inverse relation between hypocalcemia and disease activity in SLE patients.

3. Renal markers in RA: 1. Any clinical data available in these RA patients who had high creatinine, BUN, Potassium, Calcium and Magnesium and low sodium?

2. Many studies demonstrated serum Uric acid as an independent predictor of renal disease in patients with RA, but serum uric acid has not been evaluated in this present study.

6. PLOS authors have the option to publish the peer review history of their article (what does this mean?). If published, this will include your full peer review and any attached files.

Reviewer #1: No

Reviewer #2: No

---

## [Author Response · Author response to Decision Letter 0]

13 Apr 2023

Responses to Reviewers:

We sincerely thank the Reviewers for their insightful comments and helpful feedback, which have led to significant improvements in the manuscript. A point-by-point response is provided below. The line numbers referred to in the responses below correspond to the “Renal Markers and Autoimmunity (revised with track changes)”. 

Reviewer #1

Summary and general comments:

In the manuscript “Evaluation of Renal Markers in Systemic Autoimmune Diseases” the authors took a correlation approach to investigate the association between several renal markers and kidney disease in patients with systemic autoimmune diseases. To achieve such aim, the authors compared retrospectively 13 renal markers together with autoantibodies in 3 cohorts of patients: 1) 126 with systemic lupus erythematosus (SLE), 2) 695 with connective tissue diseases (CTDs) and 3) 3304 with rheumatoid arthritis (RA).

The results showed that:

1. Renal markers were elevated in the 3 cohorts of patients comparatively to controls.

2. Some autoantibodies such ANA, anti-Histone and anti-CCP were associated with kidney damage and lower eGFRs.

3. All the renal markers except for electrolytes were good in assessing renal involvement in patients with autoimmune diseases.

Overall, the paper is well organized and I enjoyed reading it but there are some rectifications and explanations that need to be addressed.

We thank the Reviewer for their study summary. This accurately represents the ideas that we wanted to put forth. 

Specific comments:

#1 In the text, the authors mention 4 tables that do not exist in the manuscript version submitted for review. Please provide the manuscript with the cited tables.

We uploaded the tables as a separate document. The tables have been added to the revised manuscript. (Table 1:86-87, Table 2: 154-155, Table 3: 173-174, Table 4: 188-190)

#2 The last paragraph of the introduction is a bit misleading. For instance, the authors mention 13841 SLE patients when in reality it is 126 patients and 13715 controls. And then who were these control subjects, and why did each of the 3 cohorts have its own control group?

The lines have been reworded for clarity on the study population (line numbers 73-74). This was a retrospective study of seropositive and seronegative data from individuals who were tested on the autoimmune and kidney function panels. 

#3 In the statistical analysis section, the authors did not specify the type of statistical tests used.

Two-tail student T test was performed to determine the p value. It has been incorporated in the ‘data analysis’ sub-section under ‘Materials and Methods’ (line numbers 132-133).

#4 In the results section, the authors did not detail the quantitative measurements of the different markers assessed in the patients. It would be more appropriate to compare the means or to perform non-parametric tests followed by ROC analysis to evaluate the sensitivity and specificity of the markers in detecting renal damage. Eventually, the authors could perform a logistic regression to estimate the incremental risk for each one-unit increase in each marker and build a predictive model for kidney damage.

The markers used in the study are routinely used in clinical laboratories and their sensitivities and specificities are well-known. We do plan to apply logistic regression to estimate the incremental risk for each one-unit increase in each marker in a subsequent study using a larger data set.

#5 In SLE patients, the association of anti-DNA antibodies with renal damage has been known for a long time and numerous human and mouse studies have confirmed the pathogenic role of these autoantibodies. The authors should discuss the discrepancy of their results with the literature.

We appreciate and thank the author for this insightful comment. Our results, as indicated in Table 2, showed some significant correlations which were not elaborated in the discussion section. We have added the following to the manuscript to highlight this association (line numbers 239-244).

“eGFR, Cystatin C, BUN, Calcium, and Albumin showed a significant association with seropositive females in the SLE group. Lupus nephritis is a common manifestation of SLE. Its onset is between 3-5 years after SLE onset. Lupus nephritis is more common in women. This is consistent with our results. Regular monitoring of kidney function by measuring renal markers and preventing the decline of kidney function is the primary treatment of Lupus nephritis.”

Reviewer #2

1. About leftover samples: when, on an average, were the samples processed after being leftover and is there any effect on the sample results because of the leftover time.

We did not use leftover samples. We performed a retrospective analysis of the clinical laboratory data after de-identifying samples under WIRB (work order #1-1098539-1)

2. The titre and the strength of ANA control is not mentioned anywhere in manuscript.

1:40 dilution was used for screening and was reflexed depending on the assay results (1:80, 1:160, 1:320, 1:640, and so on). This is included in the revised manuscript (line numbers 103-104). The clinical interpretation is left to the provider.

3. Renal markers in SLE: Among the 43.4% of SLE subjects, is there any clinical data about how many were having or developed nephritis or other renal complications, that way the results would be translated more effectively bedside. in various other studies, serum creatinine level remains statistically significant risk factor for developing nephritis in SLE patients, what about the serum creatinine levels in this study, was that statistically significant? is there any clinical data available about those who have low sodium levels because some studies have demonstrated low sodium level as an inflammatory marker of lupus, the current study would further add strength to that association in case. Was the mean Calcium level in female SLE subjects and high phosphate level in male SLE patients statistically significant? there are some studies which show inverse relation between hypocalcemia and disease activity in SLE patients.

We conducted a retrospective analysis using de-identified samples of individuals that visited Vibrant Clinical Lab for routine tests. The study is limited to serological markers and does not consider the clinical aspect. We recognized this as a limitation as indicated in line numbers 254-257. Serum creatinine levels were observed and the results are summarized in tables 2,3, and 4. A significant association was observed in female rheumatoid arthritis patients. Calcium levels were found to be significant in female SLE patients and phosphate levels were statistically significant in male SLE patients as indicated in table 2 (line numbers 154-155). 

4. Renal markers in RA: 1. Any clinical data available in these RA patients who had high creatinine, BUN, Potassium, Calcium and Magnesium, and low sodium?

As mentioned earlier, we conducted a retrospective analysis using de-identified samples of individuals that visited Vibrant Clinical Lab for routine tests. The study is limited to serological markers and does not consider the clinical aspect. We recognized this as a limitation and have added it to the manuscript (line numbers 254-257 )

5. Many studies demonstrated serum Uric acid as an independent predictor of renal disease in patients with RA, but serum uric acid has not been evaluated in this present study.

Serum uric acid is an independent marker for the prediction of renal disease. However, it is not part of the routine chemistries to access kidney damage in clinical laboratories. We will consider this in future studies.

---

## [Decision Letter · Decision Letter 1]

29 May 2023

Evaluation of Renal Markers in Systemic Autoimmune Diseases

PONE-D-22-31509R1

Dear Dr. Krishnamurthy,

We’re pleased to inform you that your manuscript has been judged scientifically suitable for publication and will be formally accepted for publication once it meets all outstanding technical requirements.

Kind regards,

Alessandro Granito

Academic Editor

PLOS ONE

Additional Editor Comments (optional):

Reviewers' comments:

Reviewer's Responses to Questions

**Comments to the Author**

1. If the authors have adequately addressed your comments raised in a previous round of review and you feel that this manuscript is now acceptable for publication, you may indicate that here to bypass the “Comments to the Author” section, enter your conflict of interest statement in the “Confidential to Editor” section, and submit your "Accept" recommendation.

Reviewer #1: All comments have been addressed

Reviewer #2: All comments have been addressed

2. Is the manuscript technically sound, and do the data support the conclusions?

Reviewer #1: Yes

Reviewer #2: Yes

3. Has the statistical analysis been performed appropriately and rigorously? 

Reviewer #1: Yes

Reviewer #2: Yes

4. Have the authors made all data underlying the findings in their manuscript fully available?

Reviewer #1: Yes

Reviewer #2: Yes

5. Is the manuscript presented in an intelligible fashion and written in standard English?

Reviewer #1: Yes

Reviewer #2: Yes

6. Review Comments to the Author

Reviewer #1: Thank you for the revision. All questions were adressed.

I feel that the manuscript is of sufficient quality and detail to Accept.

Reviewer #2: (No Response)

7. PLOS authors have the option to publish the peer review history of their article (what does this mean?). If published, this will include your full peer review and any attached files.

Reviewer #1: No

Reviewer #2: No
